# Implementation of a surgical ward innovation: Telemonitoring controlled by healthdot [SWITCH-trial PROTOCOL]

Friso Schonck[1], Misha Luyer[2,1], Nardo van der Meer[3,1], Arthur Bouwman[3,2], Simon Nienhuijs[2,1]*

1 Department of Surgery, Catharina hospital Eindhoven, Eindhoven, The Netherlands, 2 Department of Electrical Engineering, Signal Processing Systems, Eindhoven University of Technology, Eindhoven, The Netherlands, 3 Department of Anesthesiology, Catharina hospital Eindhoven, Eindhoven, The Netherlands

All authors contributed equally to this work.
* simon.nienhuijs@catharinaziekenhuis.nl

## Abstract

### Background

Monitoring patients' vital signs is important to detect abnormalities that may indicate a disturbed postoperative course. However, manual monitoring or so-called spot checks performed by nursing staff require a significant amount and are time-consuming. Wearable devices have been shown to be at least equally effective in collecting these data and have the possibility for continuous monitoring. This technology could enhance improvements in early warning score systems by continuous measurement or ease onward monitoring after earlier discharge. Furthermore, the implementation of wearables for patient monitoring may also decrease the workload for nursing staff. However, it is still unclear how workload is impacted after the implementation of continuous monitoring by wearables in a surgical ward.

### Methods

The level of workload for nursing staff will be assessed by the Integrated Workload Scale [IWS] in an observational study on the stepwise implementation of continuous monitoring at a surgical ward. In a 6-month period, the vital parameters of 500 oncological surgery patients will be recorded using an accelerometer [Healthdot ® Philips] which measures heart rate and respiration rate, gradually leaving out spot checks by nurses. Proctor et al.'s taxonomy of implementation outcomes is used to guide other outcomes: acceptability, appropriateness, feasibility, adoption, penetration, implementation cost and sustainability. Measurements will be performed by device performances of signal noise ratio and [suspected] adverse events, questionnaires Evidence Based Practice Attitude [EBPAS] and System Usability Scale [SUS] and a

**Data availability statement:** No datasets were generated or analysed during the current study. All relevant data from this study will be made available upon study completion.

**Funding:** No author have received specific funding. The trial has however received the Healthdots for clinical research free of charge without interest by Philips Research.

**Competing interests:** R.A. Bouwman acts as clinical consultant for Philips Research in Eindhoven, The Netherlands. This does not alter our adherence to PLOS ONE policies on sharing data and materials.

focus group information that will be processed and objectified by means of a Braun and Clarke thematic analysis.

## Discussion

Implementation is complex, especially within healthcare. While the validity of monitoring devices has been studied, their implementation in daily practice has been explored to a limited extent. This study focuses on implementation, with nurses as the primary research group.

## Aim

The aim is to investigate the implementation of continuous monitoring using a wearable device regarding efficiency and workload primarily on nursing perspective.

## Trial registration

ClinicalTrials.gov NCT05956210, Registered on 21 July 2023

## Background

Nursing staff play an essential role in patient care by monitoring patients' vital signs. Alterations in these parameters can often be found hours before a life-threatening event occurs [1]. Early warning scores [EWS] are embedded, especially in surgical wards, due to their promise to reduce adverse events and improve the outcomes of postoperative patients [2]. Manual vital sign monitoring or so-called spot checks can, however, be time-consuming and require a significant amount of effort from nurses, limiting their ability to attend to other clinical duties [3]. Additionally, these spot checks capture only vital parameters at a specific moment in time, and vital parameters in between remain unknown.

Wearable technology has emerged as a tool to record vital parameters and reduce the workload for healthcare providers in recent years. Several studies have shown wearables to be effective in monitoring vital signs, such as heart rate, respiratory rate, and blood oxygen levels [2,4,5]. Wearables can provide real-time information for healthcare providers, which can help them detect early warning signs of deteriorating health conditions and intervene appropriately. Continuous measuring potentially provides enough data for algorithms in such a way that interventions could reduce the impact of complications sooner than spot check monitoring. Last, earlier discharge becomes an option as monitoring can continue transmurally.

However, despite their potential benefits, wearables are not yet widely adopted in healthcare settings, and their efficacy in reducing the workload of nurses and improving patient outcomes is still a subject of investigation [6,7]. It has been shown that implementation of an automated EWS system on a surgical high dependency ward improved the number of complete assessments, registered vital signs, and adherence to the EWS hospital protocol [8]. In some studies, reductions in mortality

and hospital stay were also found [9–11]. The experienced reduction in workload and efficiency has been investigated to a lesser degree, while that is actually an important condition for successful implementation.

We will perform a prospective observational study on the stepwise implementation of continuous monitoring at the surgical ward and assess levels of acceptance, fidelity and workload among healthcare professionals. The objective of this study is to determine whether the use of wearables can reduce the workload for nurses in a surgical ward. We aim to investigate the efficacy of wearables in reducing the time required for manual vital sign monitoring, identify the benefits and challenges of using wearables, and explore the attitudes of nurses toward wearable technology.

## Methods

### Setting

This single-center study will be conducted in the oncological surgery ward of a large referral hospital. Approximately one thousand patients are admitted to this ward annually. Major abdominal oncological procedures, such as colorectal, esophageal, stomach, bladder and gynecological debulking procedures, are the main indications. In two frames of three months, consecutive series of all admitted adult patients will be eligible for remote continuous monitoring of vital parameters. Exclusion criteria are allergy for white plaster, implantable device [pacemaker/ICD or neurostimulator], pregnancy or breastfeeding. The wearable device [Healthdot ® Philips, see Fig 1]] is an accelerometer applied on the left side of the chest through which heart and breathing rate can be measured accurately[1,12]. Data are transferred unobstructively and automatically through a low energy [LoRa] network. Based on these two vital parameters, a Continuous Remote Early

| | STUDY PERIOD | | | | |
|---|---|---|---|---|---|
| | Enrolment | Allocation | Post-allocation | | Close-out |
| TIMEPOINT** | $-t_1$ | 0 | $t_{1-3}$ | $t_{4-5}$ | $T_6$ |
| **ENROLMENT:** | | | | | |
| **Eligibility screen** | X | | | | |
| **Informed consent** | X | | | | |
| **Allocation** | | X | | | |
| **INTERVENTIONS:** | | | | | |
| *Spot check/ (Healthdot)* | | | X (primary) | X | |
| *Healthdot/ (Spot check)* | | | X | X (primary) | |
| **ASSESSMENTS:** | | | | | |
| *Focusgroup* | | | X | X | X |
| *Questionnaires* | | X | X | X | X |
| *Data collection* | | X | X | X | X |

**Fig 1. SPIRIT schedule of enrollment, interventions and assessments.**

Warning Score [CREWS] was composed, and its accurateness equals the standard used Modified EWS [MEWS] in this setting [1]. The boundaries for daily practice were exposed on each computer used for the rounds [see Fig 2].

The recruitment has started in November 2022 and is ongoing.

### Implementation phase

An e-learning session about the functioning and application of the wearable data logger as well as interpretation of CREWS was composed and completed by the nurses. The telemonitoring device will be applied for any admission to the ward as a change in daily practice. At the beginning or during the hospital stay, information will be provided to the patients, and informed consent will be obtained for those who agree to share the data. This design was approved by the Medical Ethics Committee. In the first three months, manual spot check monitoring continues and provides a base for the MEWS system [see Fig 3]. The frequency is normally set to three times a day, which can be adjusted in both ways to the discretion of the healthcare providers. Recorded tele-data can be viewed in an application, so the influence of knowledge based on these vital parameters cannot be ruled out; however, this access will be granted to check its connectivity and become accustomed to interpretation. In a second timeframe of three months, spot checks will be omitted as a standard; however, they can be added to telemonitoring, again to the team's opinion.

### Outcomes

The objective of this study is the evaluation of implementation, defined as the usability, the degree of implementation and workload for nursing staff. The primary endpoint will be workload measured by the Integrated Workload Scale [IWS]. This questionnaire has multidimensional descriptions and colors of workload [from 'not demanding' up to 'work too demanding' in nine steps], suitable for periodical questioning and concurrent monitoring of tasks. Measurements will be performed throughout the entire study period. For secondary outcomes, Proctor et al.'s taxonomy of implementation outcomes will be used to guide acceptability, appropriateness, feasibility, adoption, penetration, implementation cost and sustainability [13]. Measurements will be performed by device performances of signal noise ratio and [suspected] adverse events, questionnaires Evidence Based Practice Attitude [EBPAS] and System Usability Scale [SUS] and a focus group information that will be processed and objectified by means of a Braun and Clarke thematic analysis [14]. A total of approximately 70 nurses working at the surgical ward will participate and complete IWS questionnaires. Ten nurses and 5 physicians will be included to participate voluntarily in the focus group for extensive evaluation of the implementation. The implementation outcomes and measurements that will be assessed are shown in Table 1.

The focus group will be established before the start of the study. During the information sessions, nurses and physicians will be approached for voluntary participation in the focus group. Nurses and physicians voluntarily participating in the focus group will sign informed consent for participation. The focus group will gather 3 times during the study period

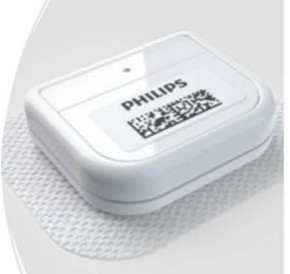

| Vital parameter score | 0 | 1 | 2 |
|---|---|---|---|
| Heart rate | 40 ≤ ≥ 110 | | <40 >110 |
| Respiratory rate | 8 ≤ ≥ 20 | <8 >20 | |

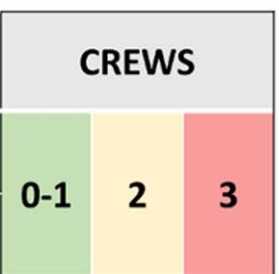

**Fig 2. Healthdot and continuous monitoring early warning score [CREWS].**

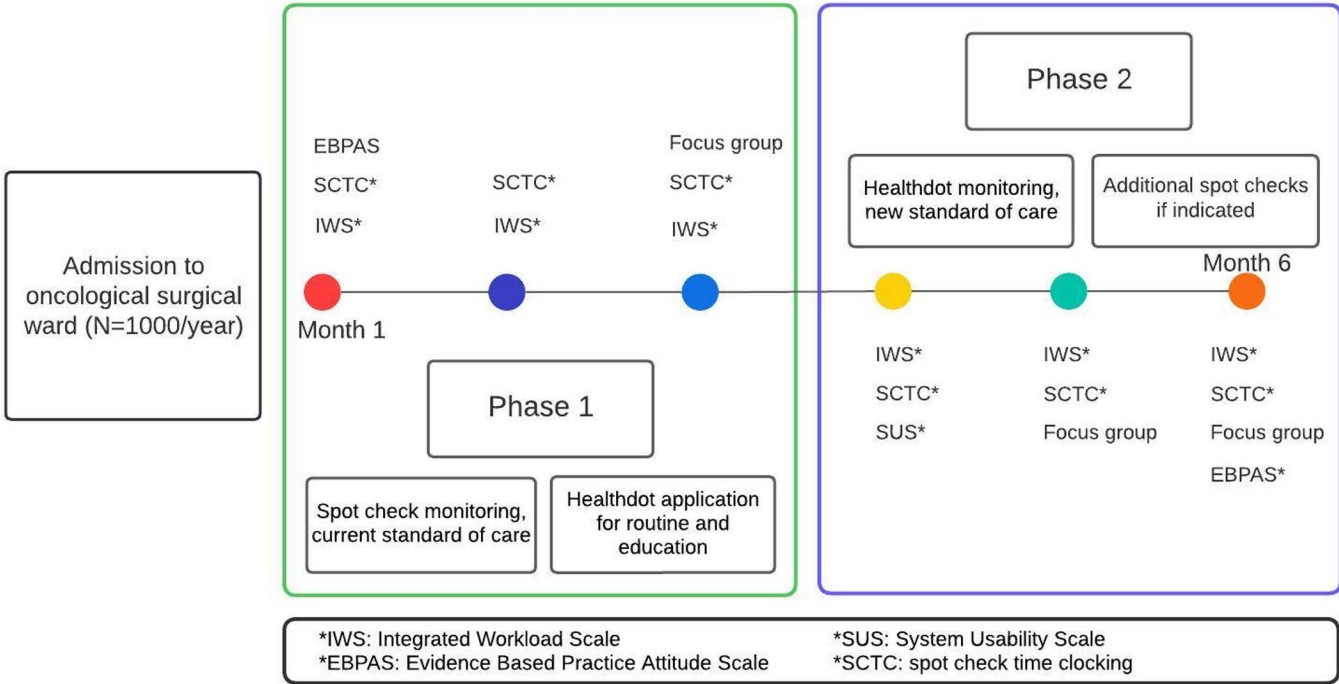

**Fig 3. Study flow chart with outcome measures.**

**Table 1. Outcomes.**

| Baseline characteristics | |
|---|---|
| Total amount of patients, n | 469 |
| Man/women, n (%) | 276 (59%)/ 192 (41%) |
| Specialization | |
| -Surgery | 276 (59%) |
| -Gynecology | 112 (24%) |
| -Urology | 75 (16%) |
| -Other | 3 (1%) |
| Mean Charlson Comorbidity Score (SD) | 5.7 (2.6) |
| Hypertension n, (%) | 122 (26%) |
| Diabetes mellitus n, (%) | 33 (7%) |
| Sleep apnea syndrome n, (%) | 33 (7%) |
| Mean duration of admission, days (SD) | 8.2 (10.6) |
| 1st re-admission n (%), Mean duration, days (SD) | 80 (17%), 6.1 (5.1) |
| 2nd re-admission n (%) Mean duration, days (SD) | 19 (4%) 3.8 (4.4) |

to share the course of the study and experiences with regard to the implementation of the Healthdot. The focus group consisting of nurses, physicians and investigators will conduct a structured analysis into the fidelpagaity, feasibility and acceptance of the implementation, and this information will be processed and objectified by means of a Braun and Clarke thematic analysis [14].

## Statistical analysis

The statistical analysis regarding the evaluation of implementation will be performed mainly by means of descriptive statistics.. Normality-based reporting will be performed using medians and interquartile ranges [IQR] or means and standard deviations [SD]. Frequencies and percentages are reported for categorical data. Each continuous parameter is checked for normality by the Shapiro–Wilk test and visually by a figure.

All analyses will be performed using IBM SPSS Statistics 26.0 for Windows [IBM Armork, New York, USA] with a 95% confidence interval and $p < 0.05$ as statistically significant.

## Discussion

Technology has been shown to catalyze change in healthcare [15]. Although smart monitoring is a hot topic in research, embedding smart monitoring in daily practice remains a challenge [16]. Implementing smart monitoring includes individual and organizational efforts with the ability to cross disciplinary, professional and cultural boundaries and collaborate interdisciplinarily [17]. Successful implementation is a prerequisite to achieve the benefits of smart monitoring with special attention to workload, the main objective of this study [17]. However, the efficacy of reducing the workload of nurses and improving patient outcomes is still elusive. Attitudes toward change, education and technical skills are factors described as potential barriers and require special attention[17]. Monitoring alarms are an essential element of continuous monitoring, as they identify potential physiological deterioration. However, alarm burden is considered a major concern in implementation, and recent studies show up to 50% false alarms, highlighting the need for optimal alarm algorithms [18].

This single-center study suggests a process theory on the adoption of smart monitoring technology and illustrates its challenges. The results of this study will contribute to the current understanding of the effective implementation of smart monitoring technology within surgical care. Research can then start building again from here.

## Supporting information

**S1 File. C1 SWITCH protocol.**
(DOCX)

## Acknowledgments

None.

## Author contributions

**Conceptualization:** Friso Schonck, Simon Nienhuijs, Arthur Bouwman, Misha Luyer.

**Supervision:** Arthur Bouwman, Misha Luyer.

**Writing – original draft:** Friso Schonck.

**Writing – review & editing:** Simon Nienhuijs, Arthur Bouwman, Misha Luyer, Nardo van der Meer.

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
