## [Decision Letter · Decision Letter 0]

31 May 2024

PONE-D-24-08777Implementation of a Surgical Ward Innovation: Telemonitoring Controlled by Healthdot (SWITCH-trial PROTOCOL)PLOS ONE

Dear Dr. Schonck,

Thank you for submitting your manuscript to PLOS ONE. After careful consideration, we feel that it has merit but does not fully meet PLOS ONE’s publication criteria as it currently stands. Therefore, we invite you to submit a revised version of the manuscript that addresses the points raised during the review process.

We look forward to receiving your revised manuscript.

Kind regards,

Emma Campbell, Ph.D

Staff Editor

PLOS ONE

On behalf of: 

Marsa Gholamzadeh, PhD

Academic Editor

PLOS ONE

Journal Requirements:

"No autor(s) have received specific funding. The trial has however received the

Healthdots for clinical research free of charge without interest by Philips Research"

6. Please ensure that you refer to Figure 3 in your text as, if accepted, production will need this reference to link the reader to the figure.

7. Please upload a copy of Figure 3, to which you refer in your text on page 5. If the figure is no longer to be included as part of the submission please remove all reference to it within the text.

8. Please include your tables as part of your main manuscript and remove the individual files. Please note that supplementary tables (should remain/ be uploaded) as separate ""supporting information"" files

Reviewers' comments:

Reviewer's Responses to Questions

**Comments to the Author**

1. Does the manuscript provide a valid rationale for the proposed study, with clearly identified and justified research questions?

Reviewer #1: Yes

Reviewer #2: Yes

Reviewer #3: No

2. Is the protocol technically sound and planned in a manner that will lead to a meaningful outcome and allow testing the stated hypotheses?

Reviewer #1: Yes

Reviewer #2: Yes

Reviewer #3: Yes

3. Is the methodology feasible and described in sufficient detail to allow the work to be replicable?

Reviewer #1: Yes

Reviewer #2: Yes

Reviewer #3: Yes

4. Have the authors described where all data underlying the findings will be made available when the study is complete?

Reviewer #1: Yes

Reviewer #2: Yes

Reviewer #3: Yes

5. Is the manuscript presented in an intelligible fashion and written in standard English?

Reviewer #1: Yes

Reviewer #2: Yes

Reviewer #3: Yes

6. Review Comments to the Author

You may also provide optional suggestions and comments to authors that they might find helpful in planning their study.

Reviewer #1: Is there any justification of the sample sizes of the nurses to complete IWS and the focus group?

If the analysis is descriptive, it is not meaningful to perform normality test and p values.

The sentence of normality test is reiterated twice within the paragraph.

Is there any analysis plan for the repeated measures of IWS?

Reviewer #2: I would like to commend the authors with their study and their clearly written trial-protocol.

There are two items that could benefit from clarification, in my opinion, which are:

- Did the nursing staff had prior experience with continuous remote monitoring? Were they given a training prior to the trial?

- Can you clarify the timeframe of the primary outcome? It now states that the IWS is measured throughout the study, but perhaps it would be wise to specifically state if that's once a month, twice-weekly, etc.

Reviewer #3: Dear authors,

thanks for the opportunity to review the interesting paper. As a nurse, I am experienced in postoperative nursing care and I am experienced in continuous vital signs monitoring using a wearable device which captures 6 vital signs (RR, SPO2, HR, PR, blood pressure and skin temperature). This background provides me with an opinion regarding implementing, using and researching continuous monitoring. I know that a lot of work still needs to done, to move forward in inhospital continuous monitoring of vital signs. That gives me the following critical notes to the study that the authors propose in this paper.

- first, I wonder why the authors choose an device that does not provide all relevant vital parameters that are suggested by the society of critical care medicine (2023) when nurses/physicians try to recognize clinical decline of patients. I would like to ask the authors to elaborate on this in the background. Furthermore, the study does not ‘asssess wearables’, but evaluate one device, so please be explicit on this in the aim and abstract.

- second, capturing vital signs is required for evaluation of clinical condition. That in itself requires effort and time, obviously; ensuring the safety of patients is fundamental to nursing care. Measuring vital signs is the basis for clinical reasoning, to interpret the condition, and set targets for treatment and evaluations with physicians/ICU team. Introducing this technology will change the way this work is done. The authors choose to evaluate the reduction workload for nurses at outcome of this change but why? Maybe it reduces workload of ICU’s by reduced IC admission, but increases workload on general wards due to earlier recognition and subsequent initiation of treatment? Do you take this time also into account? As well as using CREWS besides the regular EWS for interpretation? Is that not a point of interest: the work process instead of workload? Therefore, I would suggest to the authors to elaborate more in the introduction on the role of nurses in either vital signs measurement as well as on how the use of wearables for vital signs measurement is already integrated or used in nursing practice. That may clarify to specific need for this research question.

Thirdly, I do have some suggestions in more detail.

Abstract

-background: ‘require a significant amount’: what is required?

-wearable devices are effective on what? do you mean these are adequately able to capture vital parameters in an accurate manner?

-workload: what does workload mean, and what provokes the workload? the declining patient or manually capturing vital signs?

-unclear in the abstract what the characteristics of the device are, please explain

-implementation strategies are missing? please elaborate briefly

Background:

-authors stresses the need to reduce workload/efforts of nurses, and argue that monitoring is an option for earlier discharge by transmurally monitoring. Who is the professional that perform the surveillance of continuous monitoring either clinically or transmurally?

-clear state of science is lacking, what does systematic reviews says on devices, as well as what does studies says on effectiveness of monitoring on earlier recognition of decline? what does studies report on implementation factors, is their any knowledge on successfully adopted and well-embedded continuous monitoring?

Aim:

-plural form is used for wearables, however, only one device is evaluated

Methods

-Table 1: is visually not clear

- theory on implementation and subsequent implementation strategies are missing.

Finally, I wonder why the authors choose to report results separately, I think that results and discussion will increase the value of the study as it is involves only one wearable in one setting.

7. PLOS authors have the option to publish the peer review history of their article (what does this mean? ). If published, this will include your full peer review and any attached files.

**Do you want your identity to be public for this peer review?** For information about this choice, including consent withdrawal, please see our Privacy Policy .

Reviewer #1: No

Reviewer #2: **Yes: ** M.V. Koning

Reviewer #3: **Yes: ** Harm H.J. van Noort

---

## [Author Response · Author response to Decision Letter 1]

17 Sep 2024

The ethic approval letter from the Medical research Ethics Committees United (MEC-U) and the local approval from the Caharina hospital board is added

---

## [Decision Letter · Decision Letter 1]

8 Oct 2024

PONE-D-24-08777R1Implementation of a Surgical Ward Innovation: Telemonitoring Controlled by Healthdot (SWITCH-trial PROTOCOL)PLOS ONE

Dear Dr. Schonck,

Thank you for submitting your manuscript to PLOS ONE. After careful consideration, we feel that it has merit but does not fully meet PLOS ONE’s publication criteria as it currently stands. Therefore, we invite you to submit a revised version of the manuscript that addresses the points raised during the review process.

We look forward to receiving your revised manuscript.

Kind regards,

Marsa Gholamzadeh, PhD

Academic Editor

PLOS ONE

Journal Requirements:

Reviewers' comments:

Reviewer's Responses to Questions

**Comments to the Author**

1. Does the manuscript provide a valid rationale for the proposed study, with clearly identified and justified research questions?

Reviewer #2: Yes

Reviewer #3: Yes

2. Is the protocol technically sound and planned in a manner that will lead to a meaningful outcome and allow testing the stated hypotheses?

Reviewer #2: Yes

Reviewer #3: Yes

3. Is the methodology feasible and described in sufficient detail to allow the work to be replicable?

Reviewer #2: Yes

Reviewer #3: Yes

4. Have the authors described where all data underlying the findings will be made available when the study is complete?

Reviewer #2: Yes

Reviewer #3: Yes

5. Is the manuscript presented in an intelligible fashion and written in standard English?

Reviewer #2: Yes

Reviewer #3: Yes

6. Review Comments to the Author

You may also provide optional suggestions and comments to authors that they might find helpful in planning their study.

Reviewer #2: I have no further suggestions

Reviewer #3: - Generally, well written, well thought through reply so thank you for that

- Regarding predictive ability of respiratory and heart rate, I understand your argumentation on that. However, predictive ability is not the same as clinical utility. In that light, the recommendation to assess all the vitals for recognize clinical decline and act accordingly will maybe change the perspective on the value of good predictive ability. Maybe the authors can be explicit how the other vitals will be measured during the implementation of this device in their study?

- In the abstract, the aim now does appear before the discussion section. Unclear what the guidelines of the journal are, but this raises questions for the reader.

- I agree with the authors that the world is rapidly changing when it comes to devices. So please be clear in the abstract on what vitals the healthdot captures.

- Methods: measuring implementation outcomes based on a taxonomy does not inform about the implementation strategies. So the authors may confuse outcomes with implementation strategies which can be structured with the model of Grol & Wensing. Generally, I recognize the training for nurses as educational strategy, however it remains unclear whether dissemination, system or other strategies are used.

7. PLOS authors have the option to publish the peer review history of their article (what does this mean? ). If published, this will include your full peer review and any attached files.

**Do you want your identity to be public for this peer review?** For information about this choice, including consent withdrawal, please see our Privacy Policy .

Reviewer #2: No

Reviewer #3: No

---

## [Author Response · Author response to Decision Letter 2]

12 Dec 2024

Thank you for reviewing the article and sending the revisions. The adjustments have been made, and I look forward to hearing your feedback on the manuscript. The attached file contains the adjustments as requested.

---

## [Decision Letter · Decision Letter 2]

16 Jan 2025

PONE-D-24-08777R2Implementation of a Surgical Ward Innovation: Telemonitoring Controlled by Healthdot (SWITCH-trial PROTOCOL)PLOS ONE

Dear Dr. Schonck,

Thank you for submitting your manuscript to PLOS ONE. After careful consideration, we feel that it has merit but does not fully meet PLOS ONE’s publication criteria as it currently stands. Therefore, we invite you to submit a revised version of the manuscript that addresses the points raised during the review process.

We look forward to receiving your revised manuscript.

Kind regards,

Marsa Gholamzadeh, PhD

Academic Editor

PLOS ONE

Journal Requirements:

Reviewers' comments:

Reviewer's Responses to Questions

**Comments to the Author**

1. Does the manuscript provide a valid rationale for the proposed study, with clearly identified and justified research questions?

Reviewer #1: Yes

2. Is the protocol technically sound and planned in a manner that will lead to a meaningful outcome and allow testing the stated hypotheses?

Reviewer #1: Yes

3. Is the methodology feasible and described in sufficient detail to allow the work to be replicable?

Reviewer #1: Yes

4. Have the authors described where all data underlying the findings will be made available when the study is complete?

Reviewer #1: Yes

5. Is the manuscript presented in an intelligible fashion and written in standard English?

Reviewer #1: Yes

6. Review Comments to the Author

You may also provide optional suggestions and comments to authors that they might find helpful in planning their study.

Reviewer #1: If the analysis is descriptive, it is not meaningful to perform normality test and p values.

- Agree

The sentence of normality test is reiterated twice within the paragraph.

- Agree

These two issues were not edited in the statistical analysis section.

7. PLOS authors have the option to publish the peer review history of their article (what does this mean? ). If published, this will include your full peer review and any attached files.

**Do you want your identity to be public for this peer review?** For information about this choice, including consent withdrawal, please see our Privacy Policy .

Reviewer #1: No

---

## [Author Response · Author response to Decision Letter 3]

12 Mar 2025

Funders role has been revised in submission according to the feedback and changed for: "The funders had no role in study design, data collection and analysis, decision to publish, or preparation of the manuscript".

---

## [Decision Letter · Decision Letter 3]

24 Mar 2025

Implementation of a Surgical Ward Innovation: Telemonitoring Controlled by Healthdot (SWITCH-trial PROTOCOL)

PONE-D-24-08777R3

Dear Dr. Schonck,

We’re pleased to inform you that your manuscript has been judged scientifically suitable for publication and will be formally accepted for publication once it meets all outstanding technical requirements.

Kind regards,

Marsa Gholamzadeh, PhD

Academic Editor

PLOS ONE

Additional Editor Comments (optional):

Please only do the reviewer's comments and the minor changes requested in the final version.

Reviewers' comments:

**Comments to the Author**

Reviewer #1: In Statistical Analysis:

Delete the 2nd repeated statement "Each continuous parameter is checked for normality by the Shapiro‒Wilk test and visually by a figure."

---

## [Editor Report · Acceptance letter]

PONE-D-24-08777R3

PLOS ONE

Dear Dr. Schonck,

I'm pleased to inform you that your manuscript has been deemed suitable for publication in PLOS ONE. Congratulations! Your manuscript is now being handed over to our production team.

Kind regards,

on behalf of

Dr. Marsa Gholamzadeh

Academic Editor

PLOS ONE